# BBOPlace-Bench: Benchmarking Black-Box Optimization for Chip Placement

## Abstract

Chip placement is a crucial step in modern chip design, because it significantly impacts the subsequent process and the overall quality of the final chip. The application of black-box optimization (BBO) for chip placement has a history of several decades. Nevertheless, early attempts were hampered by immature problem modeling and inefficient algorithm design, resulting in suboptimal placement efficiency and quality compared to the more prevalent analytical methods. Recent advancements in problem modeling and BBO algorithm design have highlighted the effectiveness and efficiency of BBO, demonstrating its potential to achieve state-of-the-art results in chip placement. Despite these advancements, the field lacks a unified benchmark for thoroughly assessing various problem models and BBO algorithms. To address this gap, we propose BBOPlace-Bench, the first benchmark designed for evaluating and developing BBO algorithms specifically for chip placement tasks. BBOPlace-Bench first collects several popular tasks and standardizing their formats, thereby providing uniform and comprehensive information for optimization. Additionally, BBOPlace-Bench includes a wide range of existing BBO algorithms, including simulated annealing, evolutionary algorithms, evolution strategy, and Bayesian optimization, and evaluates their performance across different problem modelings (i.e., permutation, discrete, and mixed search spaces) using various metrics. Furthermore, BBOPlace-Bench offers a flexible framework that allows users to easily implement and test their unique algorithms. BBOPlace-Bench not only provides efficient solutions for chip placement but also expands the practical application scenarios for various BBO algorithms. The code for BBOPlace-Bench is available in the supplementary file.

## 1 Introduction

In many real-world tasks such as neural architecture search (Zoph & Le, 2017; Wang et al., 2019a;b), hyper-parameter optimization (Yao et al., 2018; Bischl et al., 2023), and chip design (Mirhoseini et al., 2020), we often need to solve black-box optimization (BBO) problems, where the objective function has no analytical form and can only be evaluated by different inputs, regarded as a "black-box" function. For example, due to the lengthy, complex, and black-box workflow of chip design, chip placement is a typical BBO problem. Besides, BBO problems are often accompanied by expensive computational costs of the evaluations, requiring a BBO algorithm to find a good solution with a small number of objective function evaluations.

Chip placement serves as a crucial process that significantly impacts the power, performance, and area (PPA) metrics of the final chip (MacMillen et al., 2000; Markov et al., 2012). A modern chip typically consists of thousands of macros (i.e., individual building blocks such as memory components) and millions of standard cells (i.e., smaller fundamental elements like logic gates). The outcome of macro placement (MP) establishes a foundational solution for subsequent processes, such as global placement (GP) and routing, thereby playing a vital role in the overall design (Tang & Yao, 2007). For instance, MP affects the placement of standard cells, and suboptimal MP results can complicate the optimal positioning of these cells, which may ultimately lead to unsatisfactory chip performance (Vashisht et al., 2020). Furthermore, inappropriate MP can result in macro blockage within the core area, adversely affecting overall chip performance by inducing issues such as routing congestion, increased wire length, and timing performance degradation (Pu et al., 2024).

Due to the black-box nature of chip placement, designers often rely on proxy metrics that can reflect the final results to guide the optimization process (Caldwell et al., 1999; Spindler & Johannes, 2007; Lu et al., 2015). One important proxy metric is half-perimeter wirelength (HPWL), which provides an approximation for the routing wirelength and is widely used to measure the placement quality (Caldwell et al., 1999; Kahng & Reda, 2006; Shahookar & Mazumder, 1991). The application of BBO for chip placement with minimizing HPWL has a history of several decades. Nevertheless, early attempts were hampered by immature problem modeling and inefficient algorithm design, resulting in suboptimal placement efficiency and quality compared to the more prevalent analytical methods (Mirhoseini et al., 2021). Recent advancements in problem modeling and BBO algorithm design have highlighted the effectiveness and efficiency of BBO, demonstrating its potential to achieve state-of-the-art results in chip placement. Despite these advancements, the field lacks a unified benchmark for thoroughly assessing various problem models and BBO algorithms.

To fill this gap, we propose BBOPlace-Bench, the first benchmark designed for evaluating and developing BBO algorithms specifically for chip placement tasks. BBOPlace-Bench first collects several popular tasks and standardizing their formats, thereby providing uniform and comprehensive information for optimization. Additionally, BBOPlace-Bench includes a wide range of existing BBO algorithms, including simulated annealing (SA) (Murata et al., 1996), evolutionary algorithms (EA) (Bäck, 1996), evolution strategy (ES) (Hansen, 2016), and Bayesian optimization (BO) (Shahriari et al., 2016), and evaluates their performance across different problem formulations (i.e., permutation for sequence pair, discrete for grid-guide, and mixed search spaces for hyperparameter optimization). We offer multiple evaluation methods, such as MP HPWL for general BBO scenarios and GP HPWL for expensive BBO scenarios. Besides, BBOPlace-Bench offers a flexible framework that allows users to easily implement and test their unique algorithms.

We conduct experiments on two popular chip datasets, ISPD 2005 (Nam et al., 2005) and ICCAD 2015 (Kim et al., 2015), comparing four algorithms: SA, EA, ES, and BO, based on three problem formulation approaches within the framework. We consider the results of the problem setup (i.e., optimizing MP HPWL and GP HPWL) and discuss the advantages and disadvantages of different modeling methods and optimization algorithms. We also compare with recently proposed RL-based methods, demonstrating the competitiveness of the BBO approach for chip placement.

BBOPlace-Bench not only provides efficient approaches for chip placement but also expands the application scenarios of BBO algorithms. Our contributions are summarized as follows:

- We propose the first benchmark in BBO for chip placement, providing an important, real-world, and challenging task for BBO algorithms. We process chip file information from different sources into a uniform structure that is easily manageable by the BBO algorithms, significantly lowering the threshold for using this problem in the BBO domain.

- We decouple problem formulation, optimization algorithms, and problem evaluation in our benchmark, making it convenient for researchers in the BBO community to compare their performance in a clear manner. Furthermore, we provide flexible definitions of problem dimensions and evaluations of different costs, facilitating the study of advanced BBO research problems such as high-dimensional optimization and expensive optimization.

- We provide extensive empirical studies and also discuss challenges and future directions for BBO in chip placement.

## 2 BACKGROUND

### 2.1 BLACK-BOX OPTIMIZATION

We consider the problem $\max_{\boldsymbol{x} \in \mathcal{X}} f(\boldsymbol{x})$, where $f$ is a black-box function and $\mathcal{X}$ is the search space, which may be discrete, continuous, and mixed. Traditional BBO algorithms are population-based search algorithms, e.g., evolutionary algorithm (EAs) (Bäck, 1996), evolution strategies (ES) (Hansen et al., 2015; Hansen, 2016), and particle swarm optimization (PSO) (Kennedy & Eberhart, 1995; Gong et al., 2015). As a type of general-purpose heuristic optimization algorithms, EAs simulate the natural evolution process with reproduction (e.g., mutation and crossover) and natural selection. They only require the solutions to be evaluated in order to perform the search, while the problem structure information, e.g., gradient information, can be unavailable, making them suitable

for BBO problems. As the mutation operators can often generate any solution in the search space, i.e., they are global search operators, EAs can converge to the global optimum (Rudolph, 1998; Zhou et al., 2019).

Bayesian optimization (BO) (Shahriari et al., 2016; Frazier, 2018) is a widely used sample-efficient method for expensive BBO problems. At each iteration, BO fits a surrogate model, typically Gaussian process (GP) (Rasmussen & Williams, 2006), to approximate the objective function, and maximizes an acquisition function to determine the next query point. Under the limited evaluation budget, traditional BO methods only have a few observations, which are, however, insufficient for constructing a precise surrogate model, leading to slow convergence. Thus, traditional BO methods struggle to effectively solve expensive BBO problems, preventing their broader applications. The basic framework of BO contains two critical components: a surrogate model and an acquisition function. GP is the most popular surrogate model. Given the sampled data points $\{(\boldsymbol{x}_i, y_i)\}_{i=1}^{t-1}$, where $y_i = f(\boldsymbol{x}_i) + \epsilon_i$ and $\epsilon_i \sim \mathcal{N}(0, \eta^2)$ is the observation noise, GP at iteration $t$ seeks to infer $f \sim \mathcal{GP}(\mu(\cdot), k(\cdot, \cdot) + \eta^2 \mathbf{I})$, specified by the mean $\mu(\cdot)$ and covariance kernel $k(\cdot, \cdot)$, where $\mathbf{I}$ is the identity matrix of size $D$. After that, an acquisition function, e.g., probability of improvement (PI) (Kushner, 1964), EI (Jones et al., 1998) or UCB (Srinivas et al., 2012), is optimized to determine the next query point $\boldsymbol{x}_t$, balancing exploration and exploitation.

However, despite BBO's practical applications in various tasks, most "real-world" scenarios in academic research are limited, primarily focusing on problems such as hyperparameter optimization of machine learning algorithms (Pineda-Arango et al., 2021; Bischl et al., 2023), neural architecture search (Ying et al., 2019), and robotic control (Todorov et al., 2012). This paper aims to formulate the important and recently popular problem of chip placement within a framework that is conducive to BBO optimization, thereby expanding the application scope of BBO. The framework we provide is user-friendly and facilitates the integration of various advanced BBO algorithms.

## 2.2 CHIP PLACEMENT

The circuit in the placement stage is considered as a graph where vertices model gates. The main input information is the netlist $\mathcal{N} = (V, E)$, where $V$ denotes the information (i.e., height and width) about all macros designated for placement on the chip, and $E$ is a hyper-graph comprised of nets $e_i \in E$, which encompasses multiple cells (including both macros and standard cells) and denotes their inter-connectivity in the routing stage. Given a netlist, a fixed canvas layout and a standard cell library, a placement method is expected to determine the appropriate physical locations of movable macros such that the total wirelength can be minimized. A chip placement solution $\boldsymbol{s} = \{(a_1, b_1), \ldots, (a_k, b_k)\}$ consists of the positions of all the cells to be placed $\{m_i\}_{i=1}^{k}$, where $k$ denotes the total number of cells. One popular objective of chip placement is to minimize the total HPWL of all the nets while satisfying the cell density constraint, which is formulated as,

$$\min_{\boldsymbol{s}} HPWL(\boldsymbol{s}) = \min_{\boldsymbol{s}} \sum_{e \in E} HPWL_e(\boldsymbol{s}), \text{ s.t. } D(\boldsymbol{s}) \leq \epsilon, \tag{1}$$

where $D$ denotes the density, $\epsilon$ is a threshold, and $HPWL_e$ is the HPWL of net $e$, which is defined as: $HPWL_e(\boldsymbol{s}) = (\max_{v_i \in e} x_i - \min_{v_i \in e} x_i) + (\max_{v_i \in e} y_i - \min_{v_i \in e} y_i)$.

There are three mainstream placement methods, i.e., analytical methods, learning-based methods, and black-box optimization methods. Analytical methods (Chang et al., 2009) place macros and standard cells simultaneously, which can be roughly categorized into quadratic placement and non-linear placement. Quadratic placement (He et al., 2013; Lin et al., 2015) iterates between an unconstrained quadratic programming phase to minimize wirelength and a heuristic spreading phase to remove overlaps. Nonlinear placement (Chen et al., 2008; Lu et al., 2015; Cheng et al., 2018) formulates a nonlinear optimization problem and tries to directly solve it with gradient descent methods. Generally speaking, nonlinear placement can achieve better solution quality, while quadratic placement is more efficient. Recently, there has been extensive attention on GPU-accelerated non-linear placement methods. For example, DREAMPlace (Lin et al., 2020; Liao et al., 2023) transforms the non-linear placement problem in Eq. 1 into a neural network training problem, solves it by classical gradient descent and leverages GPU, enabling ultra-high parallelism and acceleration and producing state-of-the-art analytical placement quality.

Learning-based approaches, particularly reinforcement learning, are a popular topic in recent chip placement discussions. GraphPlace (Mirhoseini et al., 2021) first models chip placement as a RL

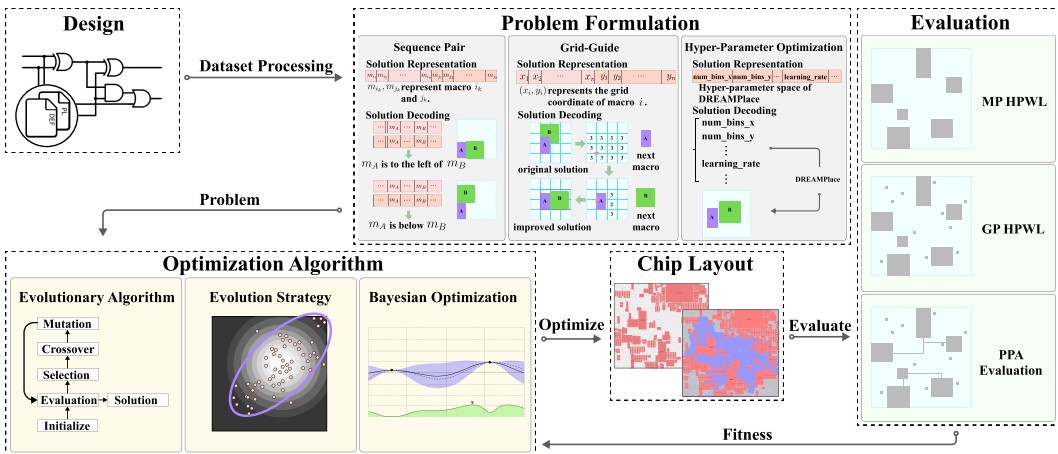

Figure 1: Illustration of BBOPlace-Bench

problem, which divides the chip canvas into discrete grids, with each macro assigned discrete coordinates of grids, wherein the agent decides the placement of the current macro at each step. Since then, many works on RL for chip placement have been proposed (Cheng & Yan, 2021; Cheng et al., 2022; Lai et al., 2022), and recent state-of-the-art works (Lai et al., 2023; Geng et al., 2024) show competitive performance compared to traditional analytical placers.

Black-box optimization methods for placement have a long history. However, earlier methods such as SP (Murata et al., 1996; Oh et al., 2022) have poor scalability due to the inefficient rectangular packing formulation. Recently, some black-box optimization methods have made significant progress by changing the search space. AutoDMP (Agnesina et al., 2023) improves DREAMPlace by using Bayesian optimization to explore the configuration space and shows remarkable performance on multiple benchmarks. WireMask-BBO (Shi et al., 2023) is a recently proposed chip placement method, which adopts a wire-mask-guided greedy genotype-phenotype mapping and can be equipped with any BBO algorithm, demonstrating the superior performance over packing-based, reinforcement learning, and analytical methods. In this paper, our proposed BBOPlace-Bench integrate these BBO problem formulation approaches into a unified benchmark for easier comparison and subsequent development of BBO algorithms for chip placement.

Recently, there have been additional benchmarks for AI in EDA. CircuitNet (Chai et al., 2023; Xun et al., 2024) focuses on providing multi-modal data for prediction tasks, enhancing the capability for various prediction tasks through the use of diverse data modalities. ChiPBench (Wang et al., 2024) emphasizes the entire EDA workflow, supplying complete files for each case and necessary design kits, thereby offering a comprehensive dataset that supports all stages of design and promotes a more integrated approach to chip design and evaluation. In contrast, our proposed BBOPlace-Bench aims to provide a unified and user-friendly benchmark for BBO in chip placement, encouraging the expansion of BBO applications in this emerging field.

## 3 BBOPLACE-BENCH

We introduce our BBOPlace-Bench in this section. The overview of our benchmark is shown in Figure 1. We first introduce how to bridge existing chip placement benchmarks, i.e., ISPD 2005 (Nam et al., 2005) and ICCAD 2015 (Kim et al., 2015), with BBO in Section 3.1. Then, we introduce the problem formulation, optimization algorithm, and evaluation in our benchmark in Sections 3.2, 3.3, and 3.4, respectively.

### 3.1 BRIDGING CHIP PLACEMENT AND BBO

With the fast development of EDA, datasets for chip design have undergone significant changes in structure and format. Early datasets, such as the ISPD 2005 (Nam et al., 2005), used a simplified *Bookshelf* format, which, however, is not suitable for real-world chip design and manufacturing due

to the significant amount of missing important information. In contrast, later datasets such as ICCAD 2015 (Kim et al., 2015) offer *LEF/DEF* format along with other necessary files, including essential information for subsequent design stages. However, these newer datasets are much more complex and contain a large amount of information that is hard to use or even unnecessary for the placement stage. To address the challenges posed by different dataset formats, we provide interfaces that are compatible with both *Bookshelf* and *LEF/DEF* formats and capable of processing them. Based on this, we extract the essential information needed for the placement stage, creating a search space that can be readily optimized with BBO algorithms. The search space in our BBOPlace-Bench can accommodate various types of search spaces, such as discrete, continuous, and mixed, which facilitates the incorporation of multiple problem formulation approaches within our framework. Specific details will be provided in the following section.

## 3.2 Problem Formulation

This section will introduce three problem formulation approaches of BBO for chip placement, where the search space sizes of SP and GG are related to the number of macros, while the search space for HPO is unrelated to it. In our BBOPlace-Benchmark, the number of macros can be specified arbitrarily and can be used for research on high-dimensional BBO, which is a recent popular topic in BBO.

**Sequence pair (SP)** is a traditional combinatorial problem formulation in chip placement (Murata et al., 1996). For $k$ macros $\{m_i\}_{i=1}^k$ to be placed, an SP is a pair of permutations of length $k$, from which the relative relationships of each macro can be extracted. Specifically, for macros $v_i$ and $m_j$, there are four relative relationships in the two permutations: $i > j$ and $i > j$, $j > i$ and $j > i$; $i > j$ and $j > i$; $j > i$ and $i > j$, which represent $m_i$ being to the left, right, above, and below $m_j$, respectively. We can use longest common subsequence to convert the SP representation to a chip placement result, which ensures minimal area placement, where no further vertical or horizontal adjustment of any macro is possible (Murata et al., 1996).

**Grid-guide (GG)** aims to directly optimize the coordinates of macros. A chip placement solution $s$ is directly represented by the coordinates of all macros $\{m_i\}_{i=1}^k$, i.e., $s = (a_1, b_1, \ldots, a_k, b_k)$, where $(a_i, b_i)$ denotes the coordinates of the macro $m_i$ on the chip canvas. However, if optimizing in this coordinates search space directly, it is difficult to efficiently find a solution that has a small HPWL value and satisfies the non-overlapping constraint. To improve the efficiency, (Shi et al., 2023) propose a wire-mask-guided greedy procedure to transform a solution into a placement result. It first divide the chip canvas into grids ($224 \times 224$ in our experiments) and determine the placement order of all the macros by some predefined rules, e.g., the area of the macro. Then, it places each macro sequentially by minimizing the incremental HPWL value based on a wire mask (Lai et al., 2022), which not only ensures a good quality of the final placement result but also avoids overlapping.

**Hyperparameter optimization (HPO)** is another problem formulation. The representative analytical method, DREAMPlace (Lin et al., 2020; Gu et al., 2020; Liao et al., 2023), performs well on many modern chips and achieves competitive results with advanced commercial EDA tools as an open-source tool. However, it has many hyperparameters that significantly affect its performance (Agnesina et al., 2023). BBO algorithms have been proven to be efficient methods for HPO and have achieved excellent performance on various tasks. In our HPO's problem formulation, we set the search space for chip placement as the hyperparameter space of DREAMPlace, with specific details shown in the Table 1. The first part consists of the general placement configurations, and the second part includes the configurations at each DREAMPlace iteration.

## 3.3 Optimization Algorithm

We use the following four typical BBO algorithms in our BBOPlace-Bench:

- Simulated annealing (SA) is a classic approach in chip placement (Murata et al., 1996). By mimicking the cooling process of metals, it effectively explores the search space, balancing exploration and exploitation to minimize objective function. Its ability to escape local minima makes it particularly valuable in optimizing complex layouts.
- Evolutionary algorithm (EA) is a population-based search framework (Bäck, 1996). We implement various operators to handle different types of search spaces. In this paper, we

Table 1: Search space of HPO in BBOPlace-Bench.

| HPO search space | Type | Range |
|---|---|---|
| GP_num_bins_x | discrete | [1024, 2048] |
| GP_num_bins_y | discrete | [1024, 2048] |
| GP_optimizer | discrete | ["adam", "nesterov"] |
| GP_wirelength | discrete | ["weighted_average", "logsumexp"] |
| GP_learning_rate | continuous | [0.001, 0.01] |
| GP_Llambda_density_weight_iteration | continuous | [1, 3] |
| GP_Lsub_iteration | continuous | [1, 3] |
| GP_learning_rate_decay | continuous | [0.99, 1.0] |
| stop_overflow | continuous | [0.06, 0.1] |
| target_density | continuous | [0.8, 1.2] |
| RePlAce_LOWER_PCOF | continuous | [0.9, 0.99] |
| RePlAce_UPPER_PCOF | continuous | [1.02, 1.15] |
| RePlAce_ref_hpwl | continuous | [150000, 550000] |
| density_weight | continuous | [1e-6, 1e-4] |
| gamma | continuous | [1, 4] |

treat SA as a specific instance of EA, utilizing a population size of one while employing the same mutation operator.

- Evolution strategy (ES) is a representative method used in the field of continuous space BBO. We integrate pycma[1], a popular implementation of CMA-ES (Hansen, 2016) in Python, into our benchmark. It not only provides a basic implementation of CMA-ES but also includes numerous advanced features suitable for high-dimensional optimization and many other scenarios.

- Bayesian optimization (BO). BBOPlace-Bench integrates one of the most popular BO frameworks, BoTorch (Balandat et al., 2020)[2]. BoTorch leverage GPUs for efficient GP fitting and inference and it includes with a wide range of advanced BO algorithms.

## 3.4 EVALUATION

As an important part of the EDA process, chip placement has many evaluation approaches. In our benchmark, we propose the following three methods.

**Macro Placement HPWL** Traditionally, the chip placement problem can be divided into two successive stages (Agnesina et al., 2023): macro placement (MP) and global placement (GP, which is also known as standard cell placement). MP heavily influences the subsequent placement of standard cells, and poor MP might make it challenging to place these cells optimally, leading to an unsatisfactory chip performance. Therefore, MP HPWL is an important metric for evaluating the quality of chip placement. Additionally, since the number of macros is much smaller compared to the number of standard cells (hundreds vs millions), MP HPWL is more suitable as an appropriate metric, especially for SP and GG formulation, which directly optimize the placement coordinates of macros.

**Global Placement HPWL** The calculation of GP HPWL is based on both macros and standard cells, and compared to MP HPWL, it is more closely related to the final chip performance. In BBOPlace-Bench, after obtaining the positions of the macros through different problem formulations and optimization algorithms, if GP HPWL evaluation is required, we will fix the already placed macros and place standard cells by DREAMPlace (Lin et al., 2020) to obtain GP HPWL, i.e., HPWL involving both macros and standard cells. Compared to MP HPWL, GP HPWL considers the total wirelength, typically on a scale that is two orders of magnitude larger, providing a better estimation of the final real performance of the final chip. Additionally, the GP HPWL interface can

---

[1] https://github.com/CMA-ES/pycma
[2] https://github.com/pytorch/botorch

also be called independently, in which case the problem is treated as an expensive BBO problem. The time overhead for different problem modeling, optimization algorithms, and evaluation methods is shown in Appendix B.1.

**PPA Evaluation** The whole chip design process is lengthy and complex, and proxy metrics (e.g., MP HPWL and GP HPWL) may not accurately capture the true performance, i.e., power, performance, and area (PPA) metrics of the chip. After obtaining the global placement results, we use commercial tool *Cadence Innovus* to proceed the subsequent stages and evaluate their PPA metrics, including routed wirelength, routed vertical and horizontal congestion overflow, worst negative slack, total negative slack, and the number of violation points. These metrics are extremely important measures of chip design and are typically considered to evaluate the quality of a chip comprehensively.

## 3.5 BBO User-Friendly Interfaces

Our proposed BBOPlace-Bench has easy-to-use interfaces, making it very easy to set up the execution of both built-in algorithms and user-customized algorithms. A simple example of running BO with GG formulation on superblue1 is shown in Code Example 1. Additionally, we provide a visualization interface that conveniently displays the placement of components, allowing for an intuitive assessment of the placement results, as shown in the Appendix B.3.

```python
from types import SimpleNamespace
from placedb import PlaceDB
from placer import REGISTRY as PLACER_REGISTRY
from algorithm import REGISTRY as ALGO_REGISTRY
args = {
    "benchmark" : "superblue1", # set chips
    "placer" : "grid_guide", # choose problem formulation
    "algorithm" : "bo", # choose optimization algorithm
    "eval_gp_hpwl" : True, # set problem formulation
    "max_evals" : 100, # set max number of evaluations
}
args = SimpleNamespace(**args)
# read chip information
placedb = PlaceDB(args=args)
# initialize placer
placer = PLACER_REGISTRY[args.placer.lower()](args=args, placedb=placedb)
# initialize bbo algorithm
runner = ALGO_REGISTRY[args.algorithm.lower()](args=args, placer=placer)
# run it!
results = runner.run()
```

Code Example 1: Run BO with GG formulation on superblue1 of ICCAD 2015.

## 4 Experiment

### 4.1 Experimental Settings

We empirically test methods in BBOPlace-Bench on the ISPD 2005 (Nam et al., 2005) and ICCAD 2015 benchmarks (Kim et al., 2015). Their detailed statistics are provided in Table 6 of Appendix A. For ISPD 2005, we use the number of macros specified in the dataset as our macros. For ICCAD 2015, since it does not specify macros, we define the largest 512 cells by area as macros. We compare three problem formulation approaches in the benchmark, evaluating multiple algorithms under each approach. Due to the enormous permutation search space of SP, BO and CMA-ES are difficult to apply; therefore, we only run SP-SA and SP-EA. Due to the difficulty of handling mixed spaces, we continuous the search space of HPO in our experiments. We conduct experiments of MP and GP evaluation on both benchmarks, with MP having 10,000 evaluation instances, while GP HPWL is set to 200 due to longer evaluation times. For methods that are particularly time-consuming (such as BO), we reduce the number of evaluations to ensure they could complete within 24 hours. All experiments are conducted using five seeds. Detailed settings of different methods are provided in our supplemental files.

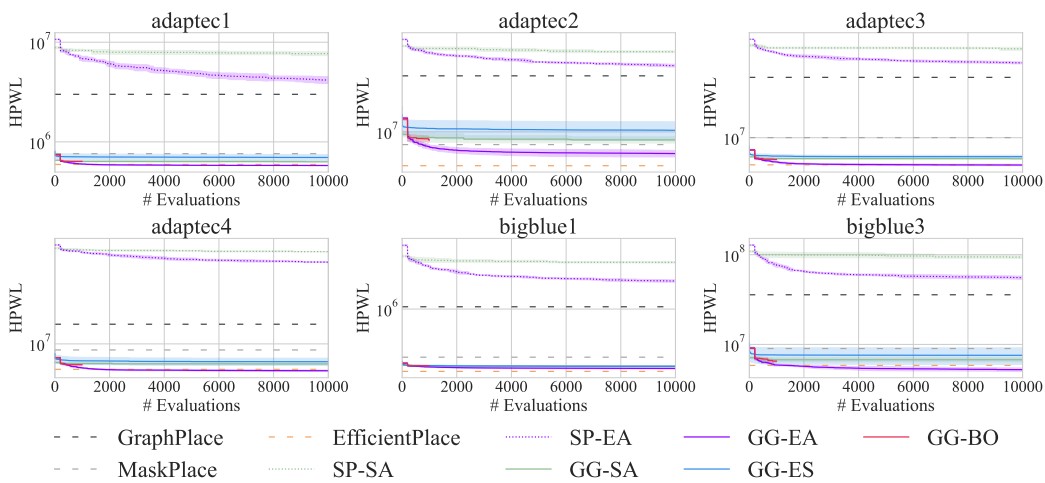

Figure 2: MP HPWL vs. number of evaluations of different methods on ISPD 2005.

Table 2: MP HPWL values ($\times 10^5$) obtained by ten compared methods on ISPD 2005. Each result consists of the mean and standard deviation of five runs. The results of three RL methods are from Geng et al. (2024). The best and runner-up methods are **bolded** and underlined, respectively. The symbols '+', '−' indicate the result is almost equivalent and inferior to the best methods, respectively, according to the Wilcoxon rank-sum test with significance level 0.05.

| Formulation | Algorithm | adaptec1 | adaptec2 | adaptec3 | adaptec4 | bigblue1 | bigblue3 | Average Rank |
|---|---|---|---|---|---|---|---|---|
| SP | SA | 76.80±3.41 - | 604.18±12.16 - | 655.61±20.30 - | 699.85±1.72 - | 31.73±0.60 - | 939.84±32.19 - | 13 |
| | EA | 41.80±3.30 - | 442.77±13.71 - | 486.91±10.24 - | 559.43±6.73 - | 20.04±0.59 - | 554.93±23.21 - | 12 |
| GG | SA | 6.32±0.05 - | 83.61±5.82 - | 64.05±0.73 - | 65.53±0.72 - | 2.44±0.02 - | 67.51±3.41 - | 6.67 |
| | EA | **5.80±0.03 +** | 61.46±4.47 - | 56.13±0.81 - | 56.79±0.80 - | 2.30±0.03 - | 52.40±2.30 - | 3.83 |
| | ES | 6.98±0.48 - | 103.66±21.81 - | 66.95±2.04 - | 68.67±5.35 - | 2.43±0.03 - | 75.66±15.96 - | 7.5 |
| | BO | 6.38±0.04 - | 83.25±3.44 - | 63.08±1.16 - | 64.34±0.86 - | 2.46±0.02 - | 65.19±2.01 - | 6.33 |
| HPO | SA | 7.89±0.12 - | 34.30±1.82 - | 53.07±0.89 + | 43.33±0.23 + | 3.45±0.07 - | 42.44±1.66 - | 4.5 |
| | EA | 7.55±0.11 - | **32.06±0.47 +** | **52.70±0.89 +** | **42.77±0.54 +** | 3.35±0.06 - | **40.03±0.72 +** | **2.83** |
| | ES | 8.15±0.12 - | 33.00±0.31 + | 53.57±0.73 + | 43.94±1.11 + | 3.40±0.04 - | 41.69±0.00 + | 4.5 |
| | BO | 9.33±0.76 - | 37.41±1.82 - | 56.27±1.59 - | 47.70±1.22 - | 3.69±0.04 - | 46.16±3.51 - | 6.17 |
| RL | GraphPlace | 30.01±2.98 - | 351.71±38.20 - | 358.18±13.95 - | 151.42±9.72 - | 10.58±1.29 - | 357.48±47.83 - | 11 |
| | MaskPlace | 7.62±0.67 - | 75.16±4.97 - | 100.24±13.54 - | 87.99±3.25 - | 3.04±0.06 - | 90.04±4.83 - | 8.33 |
| | EfficientPlace | 5.94±0.04 - | 46.79±1.60 - | 56.35±0.99 - | 58.47±1.61 - | **2.14±0.01 +** | 58.38±0.54 - | 4.33 |

## 4.2 RESULTS ON ISPD 2005

**MP HPWL comparisons.** The results on MP HPWL are shown in Figure 2 and Table 2. It can be observed that the early SP modeling has low efficiency and struggles to find satisfactory solutions, resulting in the worst performance among all algorithms. In the modeling of SP, GG, and HPO, EA consistently performs the best, demonstrating its superiority in this problem, which aligns with previous research findings (Shi et al., 2023). It is evident that BO performs relatively poorly in GG and HPO, possibly due to the large search space (i.e., 1024 dimensions), and BO's performance in high-dimensional spaces requires the assistance of additional techniques (Binois & Wycoff, 2022). Designing specific high-dimensional BO algorithms for chip placement is an interesting research question. In addition to the methods in BBOPlace-Bench, we also include three representative reinforcement learning methods as comparison methods, i.e., GraphPlace (Mirhoseini et al., 2021), MaskPlace (Lai et al., 2022), and EfficientPlace (Geng et al., 2024). These results[3] are from Geng et al. (2024). The current state-of-the-art RL method, EfficientPlace, ranks second among all methods, while HPO-EA achieves the best ranking. This demonstrates the competitive performance of BBO for chip placement across different technological approaches.

---

[3]These RL algorithms used different numbers of evaluations for training, as detailed in the original paper.

Table 3: MP HPWL values ($\times 10^4$) obtained by ten compared methods on ICCAD 2015. Each result consists of the mean and standard deviation of five runs. The best and runner-up methods are **bolded** and underlined, respectively. The symbols '+', '−' indicate the result is almost equivalent and inferior to the best methods, respectively, according to the Wilcoxon rank-sum test with significance level 0.05.

| Formulation | Algorithm | superblue1 | superblue3 | superblue4 | superblue5 | superblue7 | superblue10 | superblue16 | superblue18 | Average Rank |
|---|---|---|---|---|---|---|---|---|---|---|
| SP | SA | 12.74±0.32 - | 30.94±0.36 - | 20.81±0.59 - | 55.20±1.19 - | 24.67±0.31 - | 11.09±0.67 - | 29.75±0.43 - | 6.40±0.22 - | 10 |
|  | EA | 5.27±0.28 - | 13.34±1.04 - | 11.33±0.64 - | 31.65±2.44 - | 14.15±0.75 - | 2.31±0.19 - | 14.59±1.26 - | 2.66±0.13 - | 8.88 |
| GG | SA | 0.62±0.01 - | 1.70±0.03 - | 1.12±0.02 - | 4.16±0.07 - | 1.81±0.03 - | 0.55±0.00 - | 1.21±0.04 - | 0.53±0.01 - | 2.38 |
|  | EA | **0.59±0.00 +** | **1.55±0.01 +** | **0.95±0.01 +** | **3.84±0.03 +** | **1.72±0.02 +** | **0.54±0.00 +** | **0.95±0.01 +** | **0.49±0.00 +** | 1 |
|  | ES | 0.66±0.04 - | 1.80±0.10 - | 1.20±0.09 - | 4.78±0.37 - | 1.92±0.10 - | 0.54±0.00 + | 1.23±0.08 - | 0.53±0.02 - | 3.38 |
|  | BO | 0.63±0.01 - | 1.71±0.01 - | 1.12±0.02 - | 4.13±0.05 - | 1.84±0.03 - | 0.55±0.00 - | 1.23±0.02 - | 0.52±0.01 - | 2.63 |
| HPO | SA | 2.29±0.12 - | 4.86±0.17 - | 2.38±0.08 - | 10.45±0.23 - | 3.44±0.09 - | 1.88±0.03 - | 3.76±0.24 - | 1.56±0.15 - | 6.88 |
|  | EA | 2.07±0.14 - | 4.29±0.19 - | 2.35±0.10 - | 9.98±0.13 - | 3.19±0.03 - | 1.59±0.05 - | 3.40±0.12 - | 1.43±0.09 - | 5.13 |
|  | ES | 2.22±0.09 - | 4.83±0.24 - | 2.32±0.11 - | 10.37±0.38 - | 3.34±0.07 - | 1.69±0.10 - | 3.90±0.31 - | 1.54±0.15 - | 6 |
|  | BO | 2.60±0.07 - | 5.93±0.29 - | 2.66±0.13 - | 11.77±0.24 - | 3.93±0.09 - | 2.74±0.34 - | 4.51±0.49 - | 2.01±0.10 - | 8.13 |

Table 4: GP HPWL values ($\times 10^8$) obtained by ten compared methods on ICCAD 2015. Each result consists of the mean and standard deviation of five runs. The best and runner-up methods are **bolded** and underlined, respectively. The symbols '+', '−' indicate the result is almost equivalent and inferior to the best methods, respectively, according to the Wilcoxon rank-sum test with significance level 0.05.

| Formulation | Algorithm | superblue1 | superblue3 | superblue4 | superblue5 | superblue7 | superblue10 | superblue16 | superblue18 | Average Rank |
|---|---|---|---|---|---|---|---|---|---|---|
| SP | SA | 86.12±1.92 - | 88.37±4.59 - | 60.33±2.35 - | 109.15±5.22 - | 100.24±1.62 - | 102.07±2.31 - | 67.29±1.17 - | 30.29±0.09 - | 10 |
|  | EA | 82.03±1.82 - | 82.14±4.01 - | 56.52±2.81 - | 100.47±4.81 - | 96.41±1.99 - | 98.26±2.60 - | 65.61±1.45 - | 29.60±0.52 - | 8.88 |
| GG | SA | 62.39±0.85 - | 72.63±0.65 - | 44.58±0.66 - | 81.21±1.69 - | 84.52±0.98 - | 94.07±0.61 - | 50.39±0.55 - | 28.75±0.49 - | 6.38 |
|  | EA | 55.35±1.24 - | 62.62±1.24 - | 40.24±0.65 - | 72.15±1.16 - | 75.79±1.47 - | 88.69±1.35 - | 46.71±0.77 - | 26.96±0.15 - | 5 |
|  | ES | 66.12±2.00 - | 73.51±1.10 - | 45.58±1.13 - | 84.09±0.90 - | 85.87±2.03 - | 97.19±1.31 - | 51.47±0.48 - | 30.03±0.37 - | 8 |
|  | BO | 61.49±1.09 - | 72.65±0.99 - | 44.32±0.23 - | 81.64±0.90 - | 85.94±0.33 - | 92.12±1.52 - | 50.81±0.19 - | 29.06±0.44 - | 6.75 |
| HPO | SA | 37.51±0.73 + | 42.93±0.05 - | 28.86±0.28 - | 40.54±0.13 - | 54.26±0.49 - | 69.13±0.27 - | 37.10±0.28 + | 22.17±0.18 - | 2.75 |
|  | EA | **36.91±0.88 +** | **42.38±0.31 +** | **28.15±0.24 +** | **40.03±0.27 +** | **53.22±0.13 +** | **68.65±0.16 +** | **36.85±0.22 +** | **21.90±0.07 +** | 1 |
|  | ES | 38.04±0.14 - | 42.57±0.39 + | 28.25±0.18 + | 40.27±0.28 + | 54.55±0.75 - | 68.79±0.21 + | 37.02±0.16 + | 22.00±0.06 - | 2.25 |
|  | BO | 38.76±0.76 - | 43.95±0.42 - | 29.12±0.32 - | 41.40±0.36 - | 55.30±0.38 - | 69.85±0.49 - | 37.93±0.43 - | 22.18±0.15 - | 4 |

**GP HPWL comparisons.** The results of GP HPWL of ISPD 2005 are provided in Appendix B.2 due to space limitation. Due to its ability to comprehensively consider macros and standard cells in the layout, the advantages of HPO's GP HPWL are more pronounced.

### 4.3 RESULTS ON ICCAD 2015

**HPWL comparisons.** The results of MP HPWL and GP HPWL on ICCAD 2015 are shown in Tables 3 and 4, respectively. GG has a significant advantage in the MP HPWL task, while HPO has a notable advantage in the GP HPWL task. In both tasks, these two problem formulation approaches outperform SP. In both tasks, whether EA uses the GG or HPO problem formulation, its performance is better than BO. The convergence curves indicate that this holds true even with the same number of evaluations. This contradicts the common experience that "BO is better than EA" in many real-world tasks. This also calls for more targeted techniques for chip placement to leverage the advantages of BO in effectively utilizing optimization historical experience.

**PPA comparisons.** Due to the poor solution quality of SP, we only conducted PPA testing on the solutions modeled by GG and HPO. The PPA evaluation results of HPO modeling are shown in Table 5. The results of GG are provided in Appendix B.2 due to space limitation. For each method, we select the best chip from multiple random seeds based on GP HPWL for PPA evaluation. The chip placement is performed by different methods, and the subsequent stages and PPA evaluation are performed by *Cadence Innovus*. rWL (m) is the routed wirelength; rO-H (%) and rO-V (%) represent the routed horizontal and vertical congestion overflow, respectively; WNS (ns) is the worst negative slack; TNS (1e5 $\mu$s) is the total negative slack; NVP (1e4) is the number of violation points. WNS and TNS are the larger the better, while the other metrics are the smaller the better. From the table, it can be seen that there is currently no BBO algorithm that can dominate all algorithms. BBO for chip placement still has significant space for improvement.

Table 5: Results of PPA metrics on the ICCAD 2015 benchmarks of HPO formulation. The best and runner-up methods are **bolded** and underlined, respectively.

| Benchmark | Formulation | Algorithm | rWL | rO-H | rO-V | WNS | TNS | NVP |
|---|---|---|---|---|---|---|---|---|
| superblue1 | HPO | SA | **102.02** | 2.99 | **0.34** | **- 37.66** | **-0.37** | **0.87** |
| | | EA | 180.37 | 27.89 | 0.89 | -133.00 | -1.67 | 1.19 |
| | | ES | 152.74 | 11.72 | 0.72 | -123.60 | -1.19 | 1.10 |
| | | BO | 132.92 | **2.86** | 0.35 | -55.75 | -0.66 | 0.93 |
| superblue3 | HPO | SA | 182.25 | 27.11 | 1.51 | -134.07 | -1.12 | 0.98 |
| | | EA | 152.46 | 12.01 | 0.81 | -163.81 | -0.64 | 0.80 |
| | | ES | 155.93 | 9.75 | 0.81 | -150.05 | -0.73 | 0.78 |
| | | BO | **119.72** | **7.09** | **0.41** | **-30.07** | **-0.26** | **0.70** |
| superblue4 | HPO | SA | 101.00 | 6.54 | 0.34 | -75.61 | -1.15 | 0.95 |
| | | EA | 100.30 | 12.60 | 0.49 | -94.44 | -1.14 | 1.01 |
| | | ES | 77.07 | 1.94 | **0.11** | -25.70 | -0.47 | 0.75 |
| | | BO | **72.14** | **1.52** | 0.13 | **-24.76** | -0.42 | **0.66** |
| superblue5 | HPO | SA | 148.84 | 9.96 | 0.63 | **-58.87** | -0.59 | 1.03 |
| | | EA | **115.05** | **3.35** | **0.44** | -60.99 | **-0.27** | **0.75** |
| | | ES | 169.59 | 14.65 | 1.00 | -84.63 | -0.77 | 1.27 |
| | | BO | 129.24 | 6.58 | 0.64 | -64.06 | -0.50 | 0.82 |
| superblue7 | HPO | SA | **125.22** | **0.98** | **0.15** | -14.64 | **-0.17** | **0.86** |
| | | EA | 248.07 | 8.05 | 1.05 | -138.68 | -2.32 | 2.73 |
| | | ES | 135.24 | 4.02 | 0.42 | **-14.42** | -0.22 | 1.01 |
| | | BO | 132.42 | 2.98 | 0.34 | -16.12 | -0.24 | 1.10 |
| superblue10 | HPO | SA | 239.10 | 1.31 | 0.31 | -152.57 | -2.51 | 1.51 |
| | | EA | 196.58 | 2.41 | 0.49 | -53.69 | -1.55 | 1.46 |
| | | ES | 194.30 | 1.84 | 0.49 | -44.10 | -1.45 | 1.41 |
| | | BO | **149.47** | **0.25** | **0.05** | **-22.77** | **-0.93** | **1.30** |
| superblue16 | HPO | SA | 99.54 | 6.79 | 0.21 | -31.95 | -0.56 | 1.67 |
| | | EA | 103.18 | 2.58 | 0.23 | -52.40 | -0.63 | 1.70 |
| | | ES | 133.85 | 20.27 | 0.56 | -99.41 | -1.22 | 2.18 |
| | | BO | **76.52** | **0.27** | **0.02** | **-15.46** | **-0.30** | **1.34** |
| superblue18 | HPO | SA | 71.67 | 7.13 | 1.40 | -21.60 | -0.33 | 0.84 |
| | | EA | **57.22** | **1.92** | **0.48** | **-17.89** | **-0.25** | 0.78 |
| | | ES | 71.44 | 6.95 | 1.00 | -32.86 | -0.27 | 0.88 |
| | | BO | 73.19 | 7.73 | 1.45 | -31.34 | -0.29 | **0.76** |

## 5 CONCLUSION

In this paper, we propose BBOPlace-Bench, which is the first benchmark in BBO for chip placement. BBOPlace-Bench offers a flexible framework that allows users to easily implement and test their BBO algorithms, with the hope of facilitating the application of BBO. One limitation of this paper is that we used the commercial software *Cadence Innovus* for PPA evaluation. We plan to integrate open-source tools (e.g., OpenROAD (Kahng & Spyrou, 2021)) into our benchmark to facilitate comprehensive performance evaluation for users without a commercial license.

Based on our results, there are many worthwhile directions for future exploration. 1) **Multi-objective optimization** (Deb, 2001). In addition to the wire length considered in this paper, chip design also involves many other objectives, such as congestion and power. 2) **High-dimensional optimization** (Binois & Wycoff, 2022). Chip placement is a typical high-dimensional problem, and it is crucial to propose a targeted high-dimensional BBO algorithm based on placement characteristics. 3) **Offline optimization** (Trabucco et al., 2022) and **transfer optimization** (Bai et al., 2023). The full process of evaluating chips is expensive, but fortunately, there is a wealth of offline data available from similar chips. How to efficiently utilize this data is an interesting question.

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

## A  CHIPS STATISTICS

The detailed statistics of ISPD 2005 (Nam et al., 2005) and ICCAD 2015 contest benchmarks (Kim et al., 2015) are listed in Table 6. For ISPD 2005, we use the number of macros specified in the dataset as our macros. For ICCAD 2015, since it does not specify macros, we define the largest 512 cells by area as macros.

Table 6: Detailed statistics of the chips.

| Benchmark | # Macros | #Cells | #Nets | #Pins |
|---|---|---|---|---|
| adaptec1 | 543 | 210,904 | 221,142 | 944,053 |
| adaptec2 | 566 | 254,457 | 266,009 | 1,069,482 |
| adaptec3 | 723 | 450,927 | 466,758 | 1,875,039 |
| adaptec4 | 1329 | 494,716 | 515,951 | 1,912,420 |
| bigblue1 | 560 | 277,604 | 284,479 | 1,144,691 |
| bigblue3 | 1298 | 1,095,514 | 1,123,170 | 3,833,218 |
| superblue1 | 512 | 1,209,716 | 1215710 | 3,767,494 |
| superblue3 | 512 | 1,213,253 | 1,224,979 | 3,905,321 |
| superblue4 | 512 | 795,645 | 802,513 | 2,497,940 |
| superblue5 | 512 | 1,086,888 | 1,100,825 | 3,246,878 |
| superblue7 | 512 | 1,931,639 | 1,933,945 | 6,372,094 |
| superblue10 | 512 | 1,876,103 | 1,898,119 | 5,560,506 |
| superblue16 | 512 | 981,559 | 999,902 | 3,013,268 |
| superblue18 | 512 | 768,068 | 771,542 | 2,559,143 |

## B  ADDITIONAL ANALYSIS

### B.1  RUNTIME ANALYSIS

Due to differences in modeling approaches and optimization algorithms, the runtime varies significantly across different methods. Here, we present the average runtime for each round of different methods on adaptec3, as shown in Table 7. The fourth column of the table indicates the time taken for the algorithm search, while the fifth column shows the time taken for problem evaluation (all in seconds). It can be observed that the search duration of BO and ES is significantly longer than that of EA and SA. Additionally, the problem evaluation time for HPO is also longer than that of Grid and SP, as each run requires DREAMPlace to converge. For example, the GP evaluation time for GG is eight times that of the MP evaluation, which is due to the time overhead caused by placing many standards.

### B.2  ADDITIONAL RESULTS

In this section, we provide additional experimental results, including the curves and tables for GP HPWL on ISPD 2005, the curves for MP HPWL and GP HPWL on ICCAD 2015, as well as the PPA metrics on the ICCAD 2015 benchmarks of the GG formulation.

As shown in Table 8, the formulation of HPO combined with four BBO algorithms achieved better results than the state-of-the-art RL methods, further demonstrating the potential of BBO for chip placement. In the experiments of PPA evaluation of the GG formulation, the performance of EA and ES is generally better than BO. This is because the search space of GG is larger than that of HPO, and vanilla BO cannot demonstrate its advantages within it, as shown in Table 9.

Table 7: Runtime analysis on adaptec3.

|  |  |  | Algorithm | Evaluation |
|---|---|---|---|---|
| MP evaluation | SP | SA | 0.0271 | 0.8828 |
|  |  | EA | 0.0296 |  |
|  | GG | SA | 0.0514 | 5.6596 |
|  |  | EA | 0.1751 |  |
|  |  | ES | 1.0171 |  |
|  |  | BO | 5.3156 |  |
|  | HPO | SA | 0.0182 | 94.5737 |
|  |  | EA | 0.0494 |  |
|  |  | ES | 0.3777 |  |
|  |  | BO | 1.1098 |  |
| GP evaluation | SP | SA | 0.1496 | 43.8526 |
|  |  | EA | 0.6337 |  |
|  | GG | SA | 0.7083 | 42.7037 |
|  |  | EA | 0.6855 |  |
|  |  | ES | 1.0578 |  |
|  |  | BO | 2.6328 |  |
|  | HPO | SA | 0.1755 | 98.6505 |
|  |  | EA | 0.3815 |  |
|  |  | ES | 1.4241 |  |
|  |  | BO | 1.0724 |  |

Table 8: GP HPWL values ($\times 10^7$) obtained by ten compared methods on ISPD 2005. Each result consists of the mean and standard deviation of five runs. The results of three RL methods are from Geng et al. (2024). The best and runner-up methods are **bolded** and underlined, respectively. The symbols '+', '−' indicate the result is almost equivalent and inferior to the best methods, respectively, according to the Wilcoxon rank-sum test with significance level 0.05.

| Formulation | Algorithm | adaptec1 | adaptec2 | adaptec3 | adaptec4 | bigblue1 | bigblue3 | Average Rank |
|---|---|---|---|---|---|---|---|---|
| SP | SA | 11.87±0.28 - | 18.29±0.19 - | 31.51±0.31 - | 33.50±0.32 - | 11.54±0.12 - | 58.34±0.96 - | 12 |
|  | EA | 11.41±0.18 - | 17.37±0.26 - | 30.00±0.36 - | 33.47±0.25 - | 11.31±0.11 - | 51.98±1.62 - | 10.83 |
| GG | SA | 8.93±0.09 - | 12.08±0.50 - | 20.30±0.47 - | 21.62±0.26 - | 9.42±0.06 - | 45.09±0.99 - | 7.33 |
|  | EA | 8.49±0.08 - | 11.05±0.26 - | 18.45±0.23 - | 19.80±0.73 - | 9.29±0.05 - | 40.43±0.57 - | 6 |
|  | ES | 9.33±0.36 - | 13.39±0.58 - | 21.85±1.24 - | 23.01±0.35 - | 9.70±0.15 - | 47.31±2.21 - | 9.17 |
|  | BO | 9.01±0.20 - | 12.36±0.48 - | 20.16±0.27 - | 21.44±0.35 - | 9.45±0.03 - | 45.45±1.31 - | 7.67 |
| HPO | SA | 6.10±0.06 + | 6.95±0.12 + | 12.84±0.10 - | 12.32±0.09 - | 8.10±0.05 + | 25.36±0.77 - | 2.83 |
|  | EA | **6.05±0.03 +** | **6.82±0.08 +** | 12.73±0.11 + | **12.12±0.08 +** | **8.06±0.03 +** | **24.09±0.19 +** | 1.17 |
|  | ES | 6.09±0.03 + | 6.87±0.14 + | **12.63±0.08 +** | 12.21±0.04 + | 8.11±0.03 - | 24.72±0.60 + | 2 |
|  | BO | 6.30±0.13 - | 7.38±0.20 - | 13.01±0.10 - | 12.54±0.27 - | 8.19±0.13 + | 25.71±0.48 - | 4 |
| RL | MaskPlace | 10.86±0.18 - | 12.98±0.58 - | 26.14±0.07 - | 26.14±0.07 - | 10.64±0.01 - | 54.98±1.06 - | 10 |
|  | EfficientPlace | 7.20±0.12 - | 9.20±0.61 - | 16.49±1.07 - | 14.70±0.25 - | 8.67±0.10 - | 28.48±0.96 - | 5 |

Table 9: Results of PPA metrics on the ICCAD 2015 benchmarks of GG formulation. The best and runner-up methods are **bolded** and underlined, respectively.

| Benchmark | Formulation | Algorithm | rWL | rO-H | rO-V | WNS | TNS | NVP |
|---|---|---|---|---|---|---|---|---|
| superblue1 | GG | SA | 232.36 | 73.02 | 18.60 | -215.64 | -3.80 | 2.00 |
| | | EA | **200.27** | 62.09 | **13.62** | -252.24 | **-2.77** | **1.65** |
| | | ES | 230.76 | 79.85 | 14.95 | -195.93 | -4.09 | 2.58 |
| | | BO | 208.06 | **59.27** | 15.49 | **-117.33** | -3.47 | 2.02 |
| superblue3 | GG | SA | 262.31 | 84.39 | **10.62** | -335.84 | -3.70 | 1.57 |
| | | EA | **252.81** | **76.91** | 11.60 | -319.70 | -3.41 | **1.42** |
| | | ES | 287.73 | 100.20 | 14.18 | -317.70 | -4.00 | 1.88 |
| | | BO | 266.91 | 87.62 | 15.18 | **-210.21** | **-3.28** | 1.62 |
| superblue4 | GG | SA | 164.24 | 59.78 | 16.66 | **-107.02** | -2.58 | 1.88 |
| | | EA | 160.86 | 65.42 | 12.21 | -130.37 | **-2.57** | **1.54** |
| | | ES | **158.55** | **18.31** | **1.73** | -120.85 | -2.72 | 1.85 |
| | | BO | 160.68 | 60.89 | 19.26 | -109.85 | -2.93 | 2.02 |
| superblue5 | GG | SA | 345.06 | 30.07 | **4.23** | -299.39 | -4.06 | 2.15 |
| | | EA | **314.06** | **23.76** | 4.59 | -203.24 | **-2.65** | **1.67** |
| | | ES | 341.08 | 61.36 | 26.29 | **-160.25** | -3.09 | 1.77 |
| | | BO | 349.53 | 78.17 | 26.78 | -203.06 | -4.11 | 1.83 |
| superblue7 | GG | SA | 287.55 | **15.90** | 3.56 | -155.82 | -3.55 | 4.44 |
| | | EA | **278.62** | 16.07 | **2.47** | **-122.20** | **-3.11** | **3.89** |
| | | ES | 293.16 | 16.62 | 3.68 | -154.61 | -4.24 | 4.77 |
| | | BO | 293.97 | 21.70 | 3.95 | -122.84 | -4.46 | 4.58 |
| superblue10 | GG | SA | **391.36** | **10.60** | **2.57** | **-254.81** | **-10.20** | **2.01** |
| | | EA | 407.56 | 18.43 | 2.81 | -261.34 | -10.90 | 2.19 |
| | | ES | 427.12 | 24.54 | 4.73 | -262.63 | -11.40 | 2.09 |
| | | BO | 409.09 | 65.79 | 25.29 | -267.00 | -11.00 | 2.27 |
| superblue16 | GG | SA | 172.28 | 76.31 | **9.91** | -119.06 | -3.03 | 2.91 |
| | | EA | 164.63 | **62.06** | 11.62 | **-60.48** | **-2.79** | 2.90 |
| | | ES | **163.74** | 65.02 | 17.92 | -83.78 | -3.38 | **2.74** |
| | | BO | 184.66 | 87.94 | 11.95 | -69.18 | -3.36 | 2.91 |
| superblue18 | GG | SA | 93.70 | 10.98 | 1.64 | -51.74 | -1.23 | 1.42 |
| | | EA | 89.28 | 9.18 | 0.81 | **-40.34** | -0.66 | 1.43 |
| | | ES | **62.05** | **0.28** | **0.06** | -52.66 | **-0.56** | **1.16** |
| | | BO | 96.70 | 14.93 | 1.09 | -70.01 | -0.79 | 1.43 |

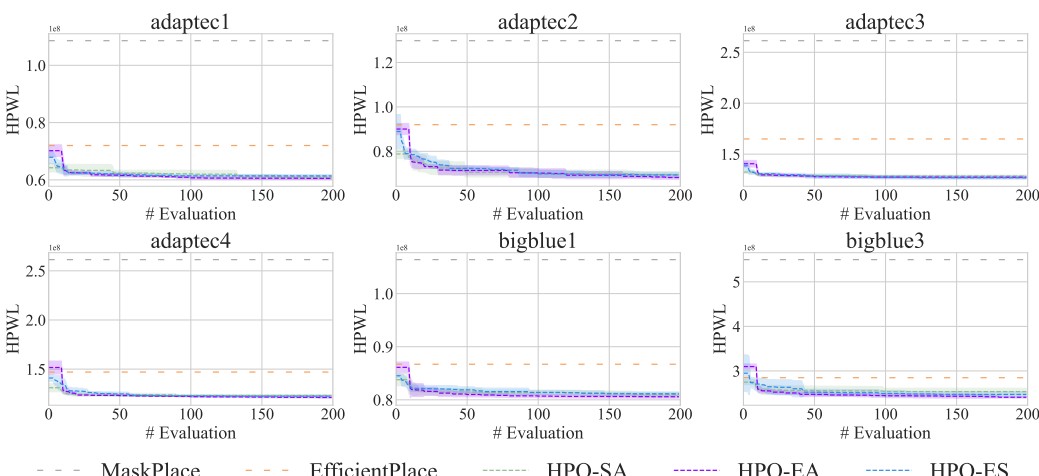

Figure 3: GP HPWL vs. number of evaluations of different methods on ISPD 2005.

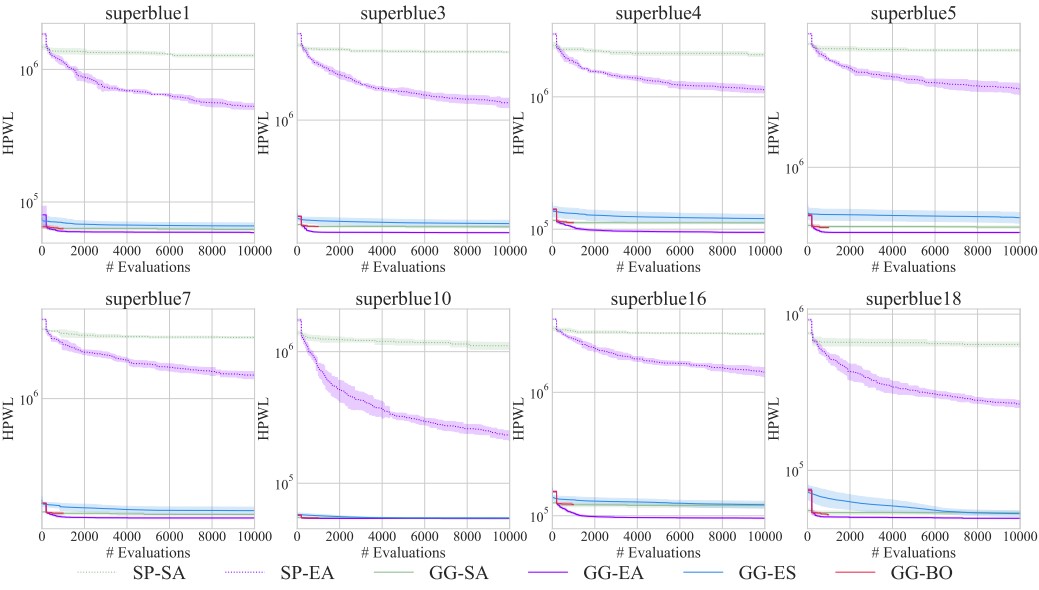

Figure 4: MP HPWL vs. number of evaluations of different methods on ICCAD 2015.

### B.3 VISUALIZATION ANALYSIS

In this section, we present the visualization results of different methods on ICCAD 2015. Our proposed BBOPlace-Bench provides a convenient visualization interface that helps users unfamiliar with the chip placement to understand the output results of their BBO algorithms.

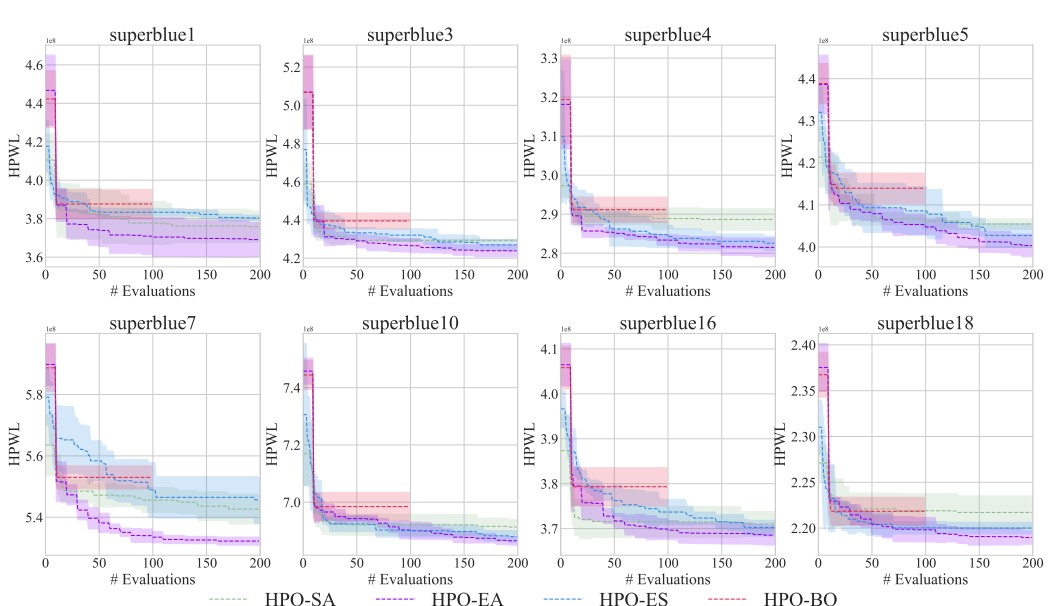

Figure 5: GP HPWL vs. number of evaluations of different methods on ICCAD 2015.

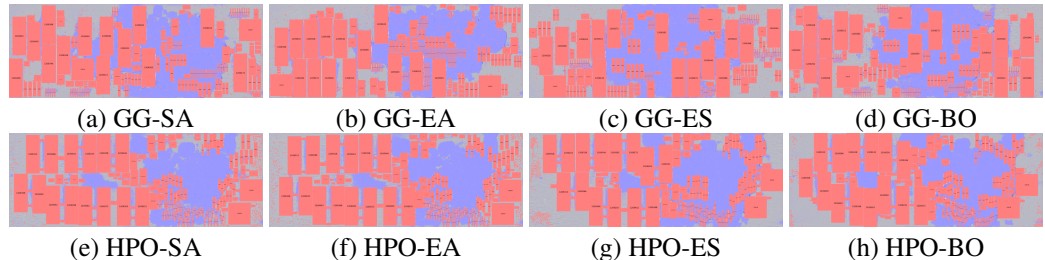

Figure 6: Visualization of superblue1.

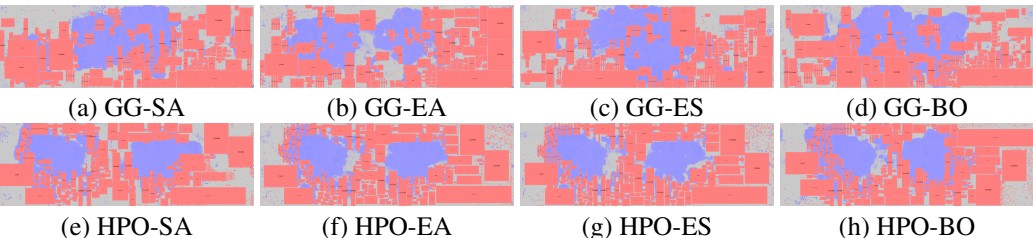

Figure 7: Visualization of superblue3.

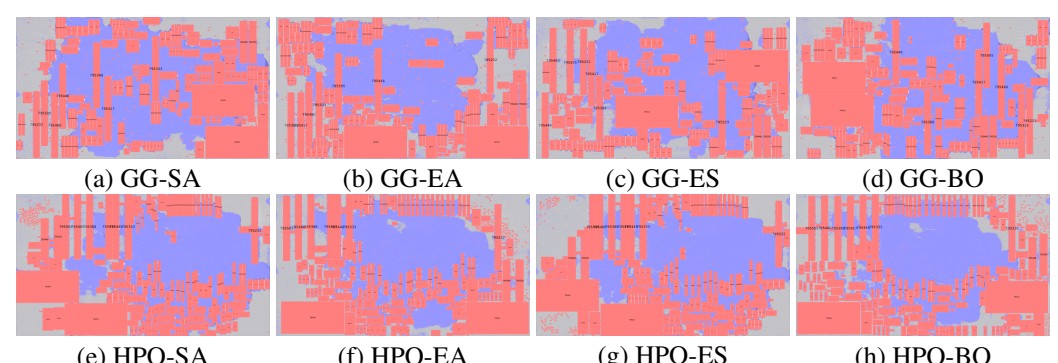

| (a) GG-SA | (b) GG-EA | (c) GG-ES | (d) GG-BO |
| (e) HPO-SA | (f) HPO-EA | (g) HPO-ES | (h) HPO-BO |

Figure 8: Visualization of superblue4.

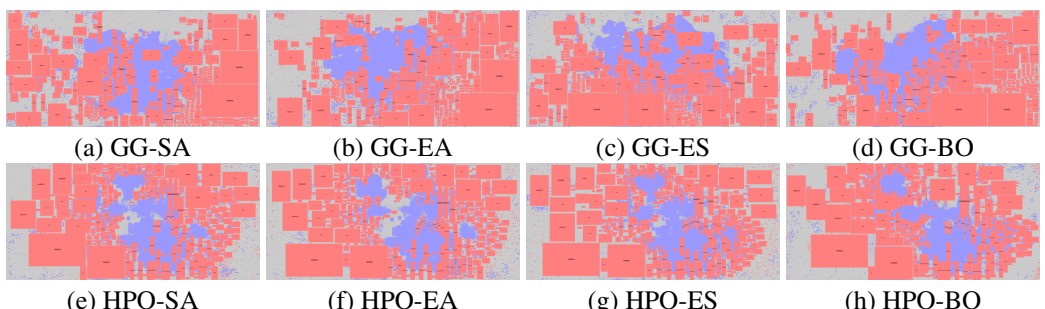

| (a) GG-SA | (b) GG-EA | (c) GG-ES | (d) GG-BO |
| (e) HPO-SA | (f) HPO-EA | (g) HPO-ES | (h) HPO-BO |

Figure 9: Visualization of superblue5.

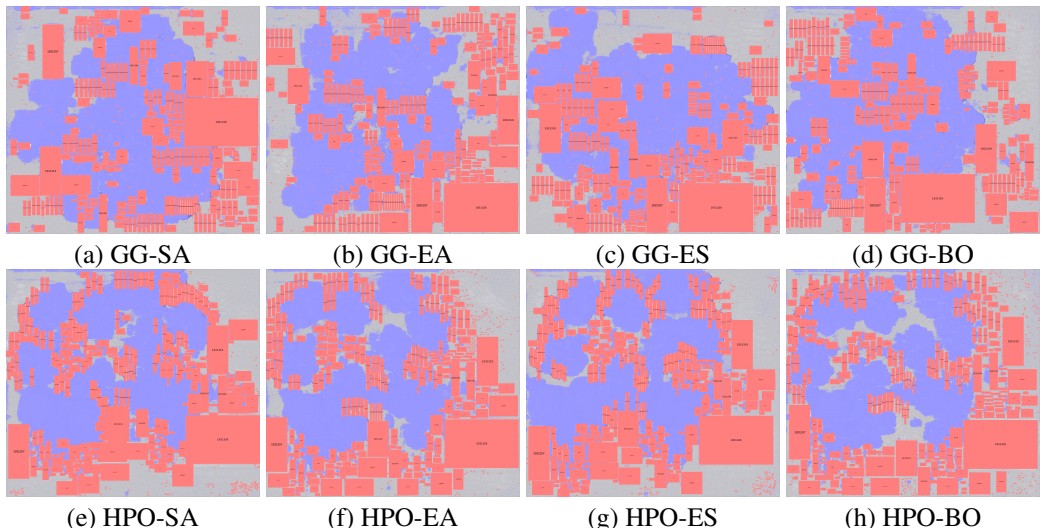

| (a) GG-SA | (b) GG-EA | (c) GG-ES | (d) GG-BO |
| (e) HPO-SA | (f) HPO-EA | (g) HPO-ES | (h) HPO-BO |

Figure 10: Visualization of superblue7.

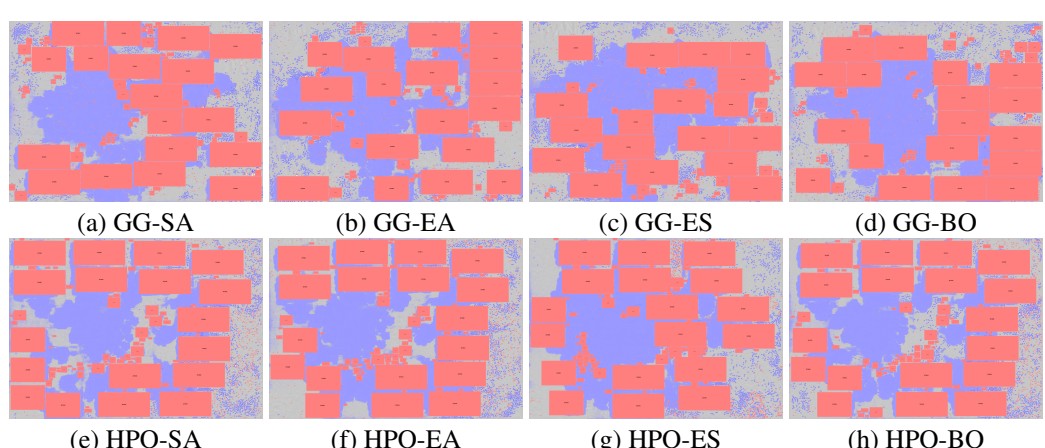

Figure 11: Visualization of superblue10.

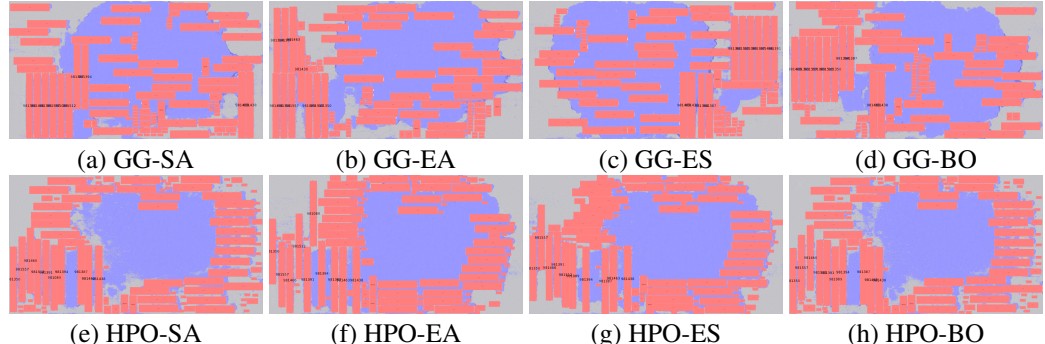

Figure 12: Visualization of superblue16.

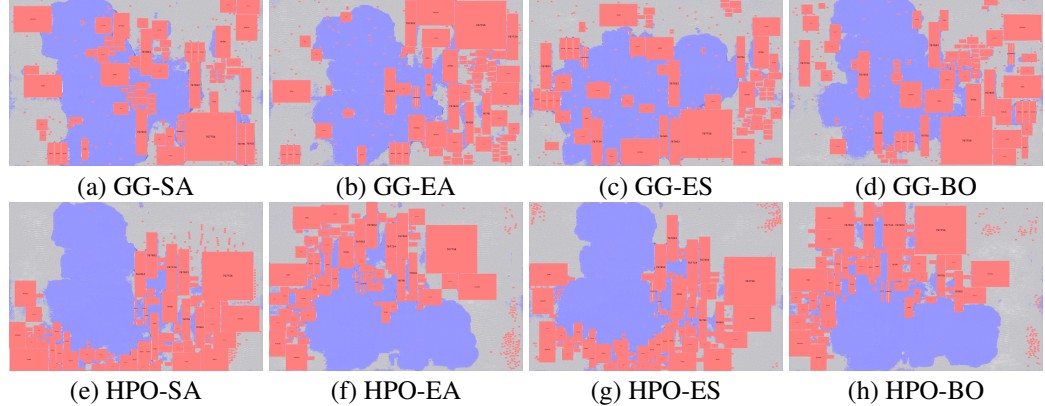

Figure 13: Visualization of superblue18.

