# OpenReview forum: "BBOPlace-Bench: Benchmarking Black-Box Optimization for Chip Placement"
_ICLR.cc/2025/Conference — ICLR 2025 Conference Withdrawn Submission_

### Official Review · Reviewer_sdKd · 2024-10-22

**Soundness:** 3
**Presentation:** 3
**Contribution:** 2
**Rating:** 5
**Confidence:** 5

**Summary:**

This paper proposes a benchmark for a black box optimization algorithm with an application scenario of ASIC placement. The execution process of the benchmark involves macro placement first, followed by standard cell placement using DREAMPlace as the backbone. The solution space for the black box optimization in the benchmark includes sequence pairs (specifying the relative order of macros), grid-guide (optimizing the coordinates of macros), and hyperparameter optimization of the DREAMPlace backbone.

**Strengths:**

- The paper is very well-written.
- The writing and illustrations are clear and help in understanding the motivation and solution.

**Weaknesses:**

- There is no comparison with analytical-based macro placement algorithms. In recent years, academia has proposed analytical-based macro placement algorithms. Additionally, there are algorithms for mixed-size placement that perform macro and standard cell placement simultaneously. Compared to RL-based methods, these algorithms are faster. The authors should include a comparison with analytical macro placement algorithms regarding performance and HPWL. This includes, but is not limited to:
  - Yuan Pu, Tinghuan Chen, Zhuolun He, Chen Bai, Haisheng Zheng, Yibo Lin, and Bei Yu. 2024. IncreMacro: Incremental Macro Placement Refinement. In Proceedings of the 2024 International Symposium on Physical Design (ISPD '24). Association for Computing Machinery, New York, NY, USA, 169–176.
  - Peiyu Liao, Dawei Guo, Zizheng Guo, Siting Liu, Yibo Lin, and Bei Yu. Dreamplace 4.0: Timing-driven placement with momentum-based net weighting and lagrangian-based refinement. IEEE Transactions on Computer-Aided Design of Integrated Circuits and Systems, 42(10):3374–3387, 2023.
  - Y. Chen, Z. Wen, Y. Liang and Y. Lin, "Stronger Mixed-Size Placement Backbone Considering Second-Order Information," *2023 IEEE/ACM International Conference on Computer Aided Design (ICCAD)*, San Francisco, CA, USA, 2023, pp. 1-9.

**Questions:**

- How is macro placement performed? Can the result of macro placement be obtained once sequence pairs and grid-guide are given?
- What is the size of the discrete solution space for SP and GG? Please provide an estimate using big O notation.
- In Section 4.1, Lines 368-370, why is it necessary to designate the largest cell as the macro? Can it be left unspecified? What impact would that have if it isn't designated?
- Can you provide references or explain how "Macro Placement HPWL" is calculated in the text?
- Why do different algorithms have different starting points in Figure 2? In terms of experimental settings, can different problem formulations like (SP and GG) have the same or as close as possible starting points? Currently, there is a tenfold difference in the starting positions in the figure.

---

### Official Review · Reviewer_jL3b · 2024-10-29

**Soundness:** 3
**Presentation:** 3
**Contribution:** 1
**Rating:** 3
**Confidence:** 4

**Summary:**

This paper discussed the background and recent development on chip placement. A number of placement benchmarks from ICCAD community are evaluated with a number of popular algorithms. A few future directions in this area are also briefly mentioned as part of the conclusion.

**Strengths:**

For those interested in chip placement, this paper could be a convenient place to see the performance of different algorithms directly compared using (presumably) popular benchmarks.

**Weaknesses:**

The comparison is billed as proposing a benchmark. This is confusing as the benchmarks are really existing benchmarks ISPD 2005 and ICCAD 2015. Beyond presenting the background and comparison results, the paper offers very little insight.

**Questions:**

Can you more explicitly explain how the proposed BBOPlace-Bench differs from or adds value beyond simply using the existing ISPD 2005 and ICCAD 2015 benchmarks? What specific contributions or modifications does BBOPlace-Bench make to create a new benchmark framework?

To those in the subfield of optimizing the placement of chips, what are some lessons from this paper that they didn't know. To those outside that specific subfield, but are interested in optimization, what is the number 1 lesson they take away from the paper?

---

### Official Review · Reviewer_D1bT · 2024-10-30

**Soundness:** 2
**Presentation:** 2
**Contribution:** 1
**Rating:** 3
**Confidence:** 5

**Summary:**

The paper introduces BBOPlace-Bench, a benchmark for evaluating and developing BBO algorithms in chip placement. BBOPlace-Bench collects several tasks and standardizes their formats. Simulating annealing, evolutionary algorithms, evolution strategy, and Bayesian optimization are included in the benchmark.

**Strengths:**

This paper presents a benchmark for chip placement that may have potential for industrial EDA applications.

**Weaknesses:**

(1)	Chip placement is an industrial problem. It is not clear that whether experiment results on the benchmark are similar to those in the real industrial scenarios. It is mentioned in subsection 4.1 that “We empirically test methods in BBOPlace-Bench on ISPD 2005 and ICCAD 2015 benchmarks”. On one hand, the reasons for choosing the ISPD 2005 benchmark and ICCAD 2015 benchmarks should be better described. On the other hand, the empirical test may not reflect the real performance in the industrial scenarios. A detailed comparison between the chosen benchmarks and real industrial scenarios is favored. Moreover, consulting with industry practitioners or comparing to recent industrial datasets on the limitations of these benchmarks and the ways to validate the benchmark's relevance to industry is suggested.

(2)	The experiments are simplified due to various reasons, which makes the results less trustable. For example, only SP-SA and SP-EA are tested and GP HPWL is set to 200. Authors should clarify the impacts of their experimental choices more explicitly. For example, if GP HPWL is set to 300 or 400, whether the experimental results will change and why. Another typical example is that why you define the largest 512 cells by area as macros for ICCAD 2015. If the number 512 is changed, whether the experimental results will change as well. If they are not easy to be demonstrated in theory, additional experiments on those parameters are suggested.

(3)	There are too many EDA references, which have little relation to do with topics of ICLR. Besides, many of the EDA references were published in the last decade or even earlier. For example, the methods in BBOPlace-Bench are tested on the ISPD 2005 and ICCAD 2015 benchmarks. Too many EDA references make the paper difficult to understand in this conference. And it is a little bit confusing that whether the problems this paper addresses are important in the current era. Clarifying the connection between these EDA references and current machine learning research relevant to ICLR is suggested. Providing more context on how these older benchmarks and references relate to current challenges in applying machine learning to chip design is also favored.

(4)	The paper has many typos or grammar issues. Even in the abstract, at least one grammar issue can be found. For example, in the abstract, “standardizes” instead of “standardizing” should be used in the sentence “BBOPlace-Bench first collects several popular tasks and standardizing their formats, …”. Another typical example is in subsection 4.1, where the word “continuous” is an adjective and cannot be used as verb in the sentence “we continuous the search space of HPO in our experiments”. The writing of this paper should be largely improved.

**Questions:**

1, The chip placement is an important process in industrial chip design. To what point, is the BBOPlace-Bench similar with the practical industrial scenarios?

2, It seems that the benchmark of HPO formulation in ICCAD 2015 has at least 18 cases. But in this paper, only 8 of them are tested and compared. Can you show the complete experimental results on the benchmark of HPO formulation in ICCAD 2015?

---

### Official Review · Reviewer_XRMg · 2024-11-03

**Soundness:** 2
**Presentation:** 3
**Contribution:** 2
**Rating:** 3
**Confidence:** 3

**Summary:**

This paper proposes BBOPlace-Bench, a benchmark for evaluating and developing a black box optimization for chip placement for EDA. BBOPlace-Bench evaluate the chip placement from problem formulation, optimization algorithm and problem evaluation, which provides a convenient tool for EDA community.

**Strengths:**

1.	This paper decouples chip placement tasks into problem formulation, optimization algorithms, and problem evaluation.
2.	This paper provides a comprehensive study of BBO for chip placement tasks.

**Weaknesses:**

1.	This paper compares traditional black-box optimization (BBO) algorithms but lacks coverage of advanced BBO methods that leverage deep learning. Although the authors include some reinforcement learning (RL)-based methods, the omission of ChiPFormer, an advanced BBO method, is notable. Including ChiPFormer would be valuable, as it introduces deep learning-based advancements that are increasingly relevant to this field. Additionally, incorporating other advanced BBO methods using deep learning would provide a more comprehensive benchmark. Specific suggestions could include recent developments in neural architecture search or multi-objective optimization approaches, which could further enrich the paper’s contributions.
2.	The paper presents evaluation results but lacks in-depth analysis of the performance differences across methods. For example, in several cases, RL-based methods perform worse than traditional BBO approaches, yet the reasons for this underperformance are not explored. To provide more insight, it would be helpful if the authors conducted a detailed analysis of these cases. They could investigate factors like differences in problem formulation, hyperparameter sensitivity, or scalability limitations that may impact RL-based methods differently from traditional BBO approaches.
3.	By focusing primarily on traditional BBO algorithms, the paper limits its contribution and does not fully capture the latest advancements in the field. To better represent the current state of BBO, the authors could consider including more recent algorithms or techniques that reflect the field’s evolution. Suggestions might include recent innovations in deep learning-based approaches like neural architecture search or multi-objective optimization techniques relevant to chip placement. Including these would enhance the benchmark’s value by offering a more complete picture of current methodologies and their practical implications in chip design optimization.
4.    After further consideration, I think the topic of this paper is more relevant with EDA conferences. Further justification of the importance of this topic and its relevance with ICLR and machine-learning are should be provided.

**Questions:**

Please see weakness.

---

### Note · Authors · 2024-11-26

I have read and agree with the venue's withdrawal policy on behalf of myself and my co-authors.